# Oleogel-Based Nanoemulsions for Beverages: Effect of Self-Assembled Fibrillar Networks on Stability and Release Properties of Emulsions

**DOI:** 10.3390/foods13050680

**Published:** 2024-02-23

**Authors:** Sai Sateesh Sagiri, Elena Poverenov

**Affiliations:** Agro-Nanotechnology and Advanced Materials Center, Institute of Postharvest and Food Sciences Agriculture Research Organization, The Volcani Center 68 HaMacabim Road, Rishon LeZion 7505101, Israel; sais@volcani.agri.gov.il

**Keywords:** oleogels, beverages, nanoemulsions, lecithin, stearic acid, sorbitan tristearate

## Abstract

Reducing the use of stabilizers is one of the main challenges in food emulsions, especially for beverages. This work aimed to produce oleogel-structured nanoemulsions (NEs) without additional surfactants. Lecithin-stearic acid (LSa) and lecithin-sorbitan tristearate (LSt) oleogels formed stable NEs under optimized sonication conditions. Microscopy and rheometry revealed that the presence of self-assembled fibrous networks (SAFiNs) in both dispersed and continuous phases provided steric stabilization to NEs. Lecithin acted as crystal habit modifier of SAFiNs and facilitated their phase partitioning. Notably, the short fibers of LSt showed better emulsifying efficiency than the long fibers of LSa. Curcumin release studies under simulated gastrointestinal conditions demonstrated that SAFiNs affect the release capabilities of NEs. Polydispersity index, zeta potential and oil syneresis data showed that the emulsions are stable for six months. Moreover, NEs showed thermal stability upon curcumin release at 25 and 50 °C. These results suggest that the developed oleogel-based NEs are suitable for the delivery of bioactive agents for beverages and other food applications.

## 1. Introduction

Beverage emulsions are unique and challenging, as they should be stable in both concentrated and diluted forms for at least six months [1,2]. Such a high stability is achieved by adding high concentrations (~20–30%) of weighting agents, and hydrocolloids are added to stabilize the dispersed oil and continuous water phases, respectively [3,4]. Brominated vegetable oil, sucrose acetate isobutyrate, ester gum and damar gum are the most commonly used weighting agents in the beverage industry [2]. In the case of hydrocolloids, amphiphilic polysaccharides such as gum arabic and modified starches (octenyl-succinated starch) are most commonly used [2,5]. Along with high amounts of stabilizers and surfactants, the addition of flavors, preservatives and colors makes beverage emulsions very complex. Reducing the surfactant and stabilizer concentration is one of the main challenges of the beverage industry. This challenge has been addressed by mixing a variety of stabilizers, synthesizing novel emulsifiers and changing process parameters [6,7,8]. However, there is room for new approaches, as the complexity of beverage emulsions still needs to be reduced.

Oleogels have emerged as structured fats that can potentially replace hard fats in bakery products, meat, dairy, etc. [9,10]. Recently, there has been a growing interest in developing novel oleogel-based emulsions for food applications [11] by using them as lipid modulators. Crystals of monoglycerides [12], wax crystals [13], modified cellulose fibers [14] and self-assembled fibrous networks (SAFiNs) of oleogelators [15] have proven their efficiency as emulsion stabilizers in spreaders [13], creams [16], salad dressings [17], etc. The wide range of oleogels and oleogel-based emulsion applications can also be attributed to their ability to release active ingredients (e.g., drugs, antioxidants, nutraceuticals) in a controlled manner [18,19]. However, these gel-based emulsions use additional surfactants and stabilizers. We are aware of only one publication in which surfactant-free oleogel-based emulsions were prepared [20]. This recent work used a homogenization of lecithin/ceramide oleogels to prepare the emulsions and studied their digestion.

Inspired by the above, we were encouraged to design oleogel-based nanoemulsions (NEs) for beverages. Previously, oleogel-based NEs were designed as gelled-oil particles using fats (monoglycerides) and composite gelators (carnauba wax/1-docosonol, ovalbumin/gum arabic) [21,22,23]. In this study, sunflower oil was structured using biocompatible gelators, i.e., lecithin (L) in combination with simple fatty acids such as stearic acid (Sa) or sorbitan tristearate (St). To our knowledge, the emulsification ability of lecithin-St (LSt) has never been explored before. The preparation of beverage emulsions traditionally involves two stages of high-pressure and high-energy homogenization. In this work, a systematic study was conducted to optimize the sonication parameters for preventing excess and insufficient homogenization. We put a significant emphasis on the stability of the prepared NEs, aiming to achieve the six-month stability required for beverage applications. We also checked the NEs’ potential as controlled delivery carriers of a bioactive agent, curcumin. In vitro release studies were conducted under simulated gastric and intestinal buffers containing pepsin and pancreatin, respectively. Since beverages are stored and transported under different conditions, the controlled release of curcumin was conducted at three temperatures (25, 37 and 50 °C).

We hypothesized that the self-assembled fibrous networks (SAFiNs) of oleogels would stabilize emulsions by structuring the dispersed oil and by increasing viscoelasticity; therefore, the emulsifying mechanism of SAFiNs could be identified. Thus, this study should bring new knowledge to the fields of lipids and colloid science and is expected to be useful for designing novel food and beverage emulsions with improved properties.

## 2. Materials and Methods

### 2.1. Materials

Analytical-grade reagents were used in this study. Soy lecithin, Sa and St were purchased from Acros Organics, Waltham, MA, USA Thermo Scientific, Waltham, MA, USA and Angene International Ltd., London, UK, respectively. The composition of soy lecithin is heterogeneous, with a total phospholipid content of 97.7 wt%, specifically 55.3% phosphatidylcholine, 25.3% phosphatidylethanolamine and 17.1% phosphatidylinositol. Na_2_HPO_4_, KH_2_PO_4_ NaCl, KCl and sodium azide were purchased from Fisher Scientific, Loughborough, UK. Pure curcumin (99%), trypsin (1:250), pepsin (3000 Unit/g) and lipase (activity eq. to 1 NF) were purchased from Sigma Aldrich, Israel. Sunflower oil was purchased from a local supermarket. Ultrapure (type 1) water was from a Merck Milli-Q Synergy^®^ UV water purification system with a resistivity of 18.2 MΩ·cm (Darmstadt, Germany).

#### Preparation of Oleogel-Based Nanoemulsions

The oleogel-based NEs were formulated by a simple two-step approach: oleogels were prepared, followed by ultrasonication. In the first step, gelators (1:1 lecithin:stearic acid (LSa) or 1:1 lecithin:sorbitan tristearate (LSt)) were solubilized in sunflower oil by heating at 80 °C and stirring at 500 rpm for 30 min. After complete dissolution, the mixture was allowed to cool to room temperature. The formation of oleogels was tested after 24 h by an inverted vial method. In the case of curcumin-loaded oleogels, stirring of the curcumin and molten gelator mixture was continued an hour before cooling.

In the second step, nanoemulsions were formed. Oleogels were pre-sheared at 1000 rpm for 10 min to achieve uniform emulsification and mixed with the aqueous phase (ultrapure water and 0.01% *w*/*w* sodium azide to prevent microbial growth) at the same speed for another 15 min to form a coarse emulsion. Then, the mixture was transferred to an ice bath and sonicated using a probe sonicator (Vibra-Cell, Sonics & Materials, Inc., Newtown, CT, USA) under pulse mode of 10 s ON and 5 s OFF. The resulting emulsion was stored at 4 °C.

The process and formulation variables were optimized as per Table 1. The acronyms for oleogels were given according to their gelator concentrations. For example, 10LSa and 10LSt represents 10% (*w*/*w*) 1:1 LSa and 1:1 LSt in sunflower oil, respectively. For NEs, the added water concentration was used. For example, 10LSaW30 denotes the emulsion of 10LSa oleogel with 30% (*w*/*w*) water.

### 2.2. Characterization of Oleogel-Based Nanoemulsions

#### 2.2.1. Emulsion Stability Studies

The stability of the oleogel-based nanoemulsions (NEs) was evaluated periodically every month by examining their physical appearance over a period of six months [24]. The dynamic stability of NEs was tested with a LUMiSizer^®^ analytical centrifuge (LUM, GmbH, Berlin, Germany). A four hundred-microliter sample was carefully loaded along the walls of the polycarbonate cuvettes to avoid air bubbles. The cuvettes were centrifuged at 3000× *g* RPM for three hours at 25 °C. The obtained data were analyzed using SEPView software, v 4.1 of the instrument.

#### 2.2.2. Average Particle Size, PDI and ζ-Potential

The average particle size (D_avg_), polydispersity index (PDI) and ζ-potential of the emulsions were determined by the dynamic light scattering (DLS) method using a Zetasizer Nano ZS90 (Malvern Instruments Ltd., Worcestershire, UK) [25]. The instrument was equipped with a He-Ne laser (Power: 4.0 mW) working at 633 nm with a backscatter detector stationed to detect a scattering angle of 173° at 25 °C. Prior to the analysis, the test samples were diluted (dilution factor: 1:4000) and then used.

#### 2.2.3. Microscopy

Light microscopy was performed on oleogels and coarse emulsions using an Olympus BX53, Tokyo, Japan. The confocal micrographs were taken using a Nikon C2 microscope (Nikon Inc., Mississauga, ON, Canada). For this, 0.01 wt% Nile red (excitation at 543 nm and detection in the range of 573–613 nm) was mixed with molecular gelators (Sa and St) and incubated at 70 °C and 200 RPM overnight in an orbital shaker incubator. The Nile red-stained gelators were then used for preparing oleogels and emulsions for confocal microscopy. The choice of Nile red was made based on the fact that it is less sensitive to high temperatures [26].

#### 2.2.4. Rheometry

A rheometer (HAAKE MARS 40, Thermo Scientific, Karlsruhe, Germany) equipped with parallel plate geometry (diameter 25 mm, gap 1 mm) was used [12]. An oscillatory shear from 0.0001 to 1 s^−1^ at a constant frequency of 1 Hz was applied on oleogels and emulsions. After finding the linear viscoelastic region (LVR) from an amplitude sweep, an oscillatory frequency sweep was carried out from 0.1 Hz to 10 Hz at a constant shear of 0.001 s^−1^. The presence of a gelator network within the NEs was verified by subjecting them to a temperature ramp from 25 to 75 °C at 5 °C/min at a constant oscillatory shear (5 rad) and frequency (0.5 Hz). The apparent viscosity of the samples was determined by increasing the rotational shear rate from 0 to 50 s^−1^ under continuous ramp mode. All the tests were carried out after equilibrating the samples at a shear of 0.0001 s^−1^ for 2 min at 25 °C.

### 2.3. Curcumin Release Studies

#### 2.3.1. Determination of Curcumin Solubility

Curcumin solubility was determined by an incremental addition method [27]. Curcumin was added with 5 mg increments to 5 g of either pure oil or molten oleogel with varied gelator proportions, at 80 °C, 500 rpm for 1 h. Solubility was assessed by checking the precipitation after centrifugation at 5000× *g* rpm for 5 min. Curcumin solubility in the supernatant was determined by measuring the absorbance at 430 nm using a UV–visible spectrophotometer (UV-1800 Shimadzu, Duisburg, Germany). The highest curcumin concentration at which neither opalescence nor precipitation was observed was noted as its maximum solubility in the tested systems.

#### 2.3.2. Determination of Curcumin Entrapment Efficiency

The amount of curcumin entrapped within the dispersed oleogel particles of the NEs was determined as previously described [28]. Briefly, samples were centrifuged at 5000× *g* rpm for 5 min. An aliquot of 30 µL supernatant was diluted in 1 mL isopropanol, vortexed for 30 s and sonicated for 10 min. After that, the sample was centrifuged at 10,000× *g* rpm for 10 min. The supernatant was collected, and curcumin absorbance was measured at 430 nm. A calibration curve of curcumin in isopropanol was generated using a serial dilution from 0.008 to 0.001 mg/mL (R^2^ = 0.997). The entrapment efficiency (*EE*) was calculated using the following equation:$$EE\;\left(\%\right)=\frac{Amount\;of\;loaded\;curcumin}{Initial\;amount\;of\;curcumin\;in\;the\;formulation}\times 100$$

#### 2.3.3. In Vitro Release Studies

The suitability of NEs as controlled carriers was studied in simulated gastric fluid (SGF) and simulated intestinal fluid (SIF) conditions [12,29]. The SGF mixture (125 mM NaCl, 7 mM KCl, 45 mM NaHCO_3_ and 0.32% pepsin) was added to the curcumin emulsion in a ratio of 3:1 and adjusted to pH 1.2 ± 0.1. The mixtures were placed in a shaking incubator (100 rpm) at 37 °C for 3 h. Further, the intestinal conditions were simulated by mixing equal volumes of gastric digested samples and SIF (10 mM phosphate-buffered solution (PBS), pH 7.2 ± 0.1, containing 1% (*w*/*v*) pancreatin) and incubation in a shaker incubator (100 rpm) at 37 °C for 6 h. Meanwhile, incubation sampling (0.5 mL) was carried out every 30 min, followed by centrifugation at 10,000× *g* rpm and 4 °C (to stop the digestion process) for 10 min, and then the released curcumin was spectrophotometrically measured at 430 nm. The effect of temperature on the release of curcumin was studied by following the same procedure in phosphate buffer, pH 7.0 at 25 and 50 °C.

### 2.4. Statistical Analysis

All experiments were performed in triplicate, and the results were reported as the mean ± standard deviation. The results were evaluated using one-way analysis of variance (ANOVA) at a significance level of 0.05 using Origin^®^ software (ver. 8.5, OriginLab Corporation, Northampton, MA, USA).

## 3. Results and Discussion

### 3.1. Preparation and Optimization of NEs

The oleogel-based NEs were formulated by a simple two-step approach, in which oleogels were prepared, followed by ultrasonication with the aqueous phase. The crystal morphology of oleogels was identified after checking their physical stability. All LSt and LSa oleogels were stable, except for 5LSt (the oleogel with 5% LSt in sunflower oil), which showed the flow of oil (Figure 1a). Light microscopy revealed large interconnected fibers and small hair-like fibers in LSa and LSt oleogels, respectively (Figure 1a). The formation of independent small fibers with no interconnections created sparse gelator networks in the LSt oleogels (insert in Figure 1a). This kind of microarchitecture seems responsible for the poorer gelation of LSt compared to LSa. Among the stable oleogels, 10LSa and 10LSt (oleogels with 10% LSa and LSt, respectively) were chosen to make NEs (Figure 1b). Once the emulsifying ability of the oleogels was identified, process conditions were optimized according to the levels given in Table 1 (details of optimization studies are given in the Appendix A). The optimized parameters to obtain stable NEs were found to be as follows: 50% sonication amplitude for 5 min, 70% *w*/*w* water fraction and 10–30% oleogel fraction (Table 2). At higher oleogel fractions, emulsions could not form because of non-uniform homogenization.

### 3.2. Determination of Emulsification Mechanism

The influence of lecithin was different on both oleogels, evident from their microstructures. For better understanding, microscopy was conducted upon controls that contained oil with pure gelators (lecithin, Sa, St). The concentration of gelators in controls was maintained identical to the concentrations in the stable 10LSa and 10LSt oleogels. The inverted vials in Figure 2 (on the left) show that lecithin and St lack the capability to structure sunflower oil, whereas Sa demonstrated this capability. Light microscopy could not identify the micellar structures of lecithin, whereas Sa oleogels showed needle-like thin fibers (Figure 2b), which is in agreement with the published literature [30,31]. In the presence of lecithin, Sa fibers’ diameter increased and formed thick, birefringent fibers (Figure 1a and Figure 2d). The effect of lecithin on St oleogels is even more pronounced. Pure St formed small crystals, appearing as dark spots (Figure 2c). In the presence of lecithin, small hair-like fibers were formed with no branching (Figure 1a and Figure 2e). This indicates that lecithin acted as a crystal habit modifier, as the size and shape of Sa and St crystals was altered. The change in gelators’ microstructure and increased gelation efficiency indicates the synergetic action of lecithin with Sa and St gelators. In addition to the role of a crystal habit modifier, lecithin performs a secondary function of promoting weak and transient junctions between the SAFiNs and water by forming hydrogen bonds [32]. These junctions create additional stabilizing networks within and around the dispersed oil with water, thereby facilitating the gelled-oil dispersions.

The emulsifying ability of the gelators was studied using confocal microscopy at each stage of emulsification, before (coarse emulsions) and after (fine emulsions) sonication. Irrespective of gelator type, both the emulsions showed sunflower oil dispersion (red) in an aqueous (dark background) medium. Either pure or in association with lecithin, St showed better emulsification, having smaller and uniform droplets in comparison to Sa. The coarse emulsions of Sa and LSa showed fibers in the water phase, indicating the phase partitioning of fibers during homogenization. Since Sa fibers are hydrophobic, in the control sample, it can be seen that some of them aggregate (indicated by arrows in Figure 2b) to minimize their contact area with the surrounding aqueous phase. Notably, in LSa-based coarse emulsions, such fiber aggregation is minor, since lecithin stabilizes LSa fibers in water by forming hydrogen bonding. In the case of the St and LSt coarse emulsions, the St fibers were too small to be seen under the microscope. This indicates that the interfacial and/or phase-partitioned self-assembled fibril networks (SAFiNs) provide steric stabilization. This type of colloidal stability was seen with the fibers of Quillaja saponin and ethyl cellulose [14,33]. Unlike Sa fibers, St fibers have polar sorbitol moiety; therefore, aggregation was not manifested. The effect of the SAFiNs was more evident after sonication, when 10LSa and 10LSt formed stable nanoemulsions, but the controls did not (vials of fine emulsions in Figure 2). In the nanoemulsions, some of the dispersed oil droplets showed fluorescence, indicating the presence of gelator fibers within them. Along with the fluorescent droplets, dark-colored oil droplets are also seen (indicated by arrows in Figure 2d,e). This could be because of the loss of fibers to the surrounding aqueous phase during sonication. As the majority of the droplets showed fluorescence, confocal microscopy confirms the retention of gelator networks in the dispersed oil.

Both SAFiNs within the oil phase and phase-partitioned SAFiNs in the aqueous phase are responsible for the stability of the emulsions. The phase-partitioned SAFiNs provide steric stabilization, and SAFiNs within the oil phase prevent phase separation by enhancing the viscoelasticity of the oil phase.

#### 3.2.1. Size and ζ-Potential of the NEs

The stability was further estimated by DLS on both fresh and diluted NEs with 70% *w*/*w* water fraction. The freshly prepared 10LSaW70 and 10LStW70 were diluted 10× and stored at 4 °C. Light microscopy showed polydispersed droplets in all the coarse emulsions (Figure 3). A notable difference is that the 10LSt emulsions had smaller droplets. DLS upon NEs showed that D_avg_ was lower in both concentrated emulsions (Figure 3e). The higher D_avg_ and PDI of diluted emulsions can be attributed to the lack of a sufficient amount of stabilizing gelators. ζ-potential values were found to be in the range of −18 to −32 mV for all the concentrated and diluted emulsions (Figure 3e). When tested after six months of storage, an insignificant decrease in the ζ-potential values (−29 to −16 mV) was observed in the NEs (Appendix A). This indicates that SAFiNs have prevented the oil droplets’ coalescence and stabilized the NEs. The insignificant change in droplet size was also evident from the D_avg_ and PDI values (Appendix A).

Confocal micrographs also confirmed the stability of stored NEs by showing distinct dispersed oil droplets (Figure 4b,d). The structural integrity of SAFiNs after storage was proved by conducting light microscopy upon the corresponding oleogels (Figure 4a,c). Their microstructure remained same as that in the fresh oleogels (Figure 1a); henceforth, NEs’ stability was maintained during the storage. Though the concentrated emulsions remained stable (Figure 4), faint phase separation was seen in the diluted 10LSaW70 emulsion (Appendix A). This indicates the weak emulsification efficiency of LSa’s large SAFiNs. Based on the obtained results, it was inferred that stable NEs for long-term use can be prepared using 10LSa and 10LSt oleogels without adding additional surfactants. This study also proves the higher emulsification efficiency of LSt’s short fibers against LSa’s long fibers.

#### 3.2.2. Oil Syneresis from NEs

The emulsifying efficiency of SAFiNs was further understood by observing the phase-separated oil (oil syneresis) from NEs. This study was conducted upon NEs with 30, 50 and 70% *w*/*w* water. The emulsions with higher oleogel fractions (10LSaW30, 10LStW30) showed the oil syneresis (Appendix A). At higher concentrations of gelators, the SAFiNs might have aggregated to minimize the interfacial tension with water, which in turn lead to the oil droplets’ coalescence and phase separation. The aggregation of SAFiNs was noticed in 10LSaW70 too (Figure 2d). This process seems to be intensified in the emulsions having a higher SAFiNs fraction. The emulsifying efficiency of SAFiNs was understood by quantifying the oil syneresis in 10LSaW30 and 10LStW30 for six months (Appendix A). Oil syneresis was higher in the first two months and then reached a plateau in both the emulsions. Higher syneresis in 10LSaW30 indicates that the long fibers of LSa have poor emulsifying efficiency compared to the short fibers of LSt.

#### 3.2.3. Dynamic Physical Stability of the NEs

The long-term stability and underlying destabilizing mechanism of NEs’ phase separation was determined using the LUMiSizer photocentrifuge. The test was conducted upon liquid NEs (10LSaW70, 10LStW70, 10LSaW90 and 10LStW90), as the transfer of thick emulsions (e.g., 10LSaW30, 10LStW30) into the LUMiSizer cuvettes was tedious. Figure 5 shows the profiles of diluted NEs (10×), as it was difficult to decipher the destabilizing phenomenon using the data obtained from the original NEs (Appendix A). The initial high transmission of near 85% due to an empty cuvette (shaded region) was dropped to almost zero at about 110 to 115 mm, indicating the top surface of the turbid emulsions. The initial profiles (red-colored bottom lines) of the emulsions remained close to zero, meaning that no light was transmitted through the tubes due to the homogeneous distribution of the dispersed oleogel in the emulsions. As the centrifugation progresses, movement of the transmission (indicated by the arrow in the 10LStW50 profile of Figure 5) towards the left of the graph was seen and ended with green-colored lines for the final transmission. This kind of transmission progression indicates creaming as the mode of the destabilization mechanism. The six-month oil syneresis study also showed this mode of phase separation.

The stability of NEs was quantified by calculating the emulsion instability index (EII) (Detloff, Sobisch and Lerche, 2013). Briefly, the EII quantifies the clarification in transmission based on particle size and separation at a given time during centrifugation divided by the maximum clarification possible (Yerramilli and Ghosh, 2017). A higher EII indicates that higher creaming occurred in 10LSa-based emulsions (Figure 5). Also, a higher EII was observed with 10LSaW90 and 10LStW90, but the differences are not significant (*p* > 0.05) compared to 10LSaW70 and 10LStW70, respectively. The higher EII of 10LSaW90 and 10LStW90 confirms the lack of a sufficient gelator network for stabilizing the dispersed phase. The increase in the thickness of the cream layer could also be due to the migration of gelator fibers along with the oil droplets. The phase separation behavior can be corroborated with the dispersed particle size distribution of the NEs. The D_avg_ of the dispersed oleogel particles in the NEs (from DLS study) was also plotted in Figure 5 and followed a similar trend as that of the EII. The accelerated or dynamic stability studies infer that LSt emulsions are more stable compared to LSa emulsions. Additionally, the emulsions with 70% *w*/*w* water tend to be more stable than the emulsions with 90% *w*/*w* water fraction.

Thus, phase-partitioned SAFiNs provided steric stabilization to NEs. The difference in the emulsification behavior of the LSa and LSt oleogels is due to the presence of different-sized SAFiNs. The short fibers of LSt are found to be effective emulsifiers compared to the long fibers of LSa. DLS studies (evident from D_avg_, PDI and ζ-potential values), oil syneresis and dynamic stability studies (evident from EII values) confirmed the emulsifying efficiency of SAFiNs.

### 3.3. Rheology Studies

#### 3.3.1. Temperature Ramp

Oleogels are thermosensitive materials, and a linear increase in temperature from 25 to 75 °C would divulge the information about the gelator networks in NEs. Both oleogels and NEs showed typical gel-like behavior by having G′ > G″ (Figure 6a,b). In the case of oleogels, a decrease in G′ began at ~45 °C. A disruption in the junction zones was initiated at this temperature, and this process continued until the complete breakdown of the gelator network. The gel-to-sol phase transition temperatures (T_gs_, the temperature at which the G′ meets G″) for 10LSa and 10LSt are 62 °C and 55 °C, respectively. The higher T_gs_ indicates that 10LSa has a higher thermal strength compared to 10LSt. The variation in T_gs_ can be attributed to the presence of more junction zones and interconnected long gelator fibers in 10LSa compared to 10LSt (evident from the oleogel micrographs in Figure 1 and Figure 2).

The observed solid-like behavior (G′ > G″) can be attributed to the gelator networks in NEs. Unlike oleogels, the viscoelastic moduli of NEs were parallel to each other for a longer duration of time. The decreasing distance between G′ and G″ curves with increasing temperature indicates the loss of a solid-like nature. The longer LVR in NEs indicates the decreased influence of temperature on SAFiNs. The lowered temperature effect is due to the random distribution of SAFiNs in the oil dispersion and also in the aqueous phase (as phase partitioned SAFiNs). Higher thermal stability of the NEs is also evident from a higher T_gs_ (~70 °C) in 10LSaW50 and 10LStW50 compared to their corresponding bulk oleogels. This kind of phase transition was not seen with the emulsions with a higher water fraction (10LSaW90 and 10LStW90 in Figure 6a,b). The gel-to-sol transition in such emulsions might have gone undetected due to the presence of a small gelator network. Though thermal strength increased, the gel strength of NEs (G′_LVR_, firmness) was decreased compared to bulk oleogels. The phase partitioning of gelators from oil droplets to the aqueous phase (seen in coarse emulsions, Figure 2) might lead to a reduction in gel strength and a shift in T_gs_ [12]. Though the strength of NEs was compromised, gelator partitioning prevented the phase separation by steric stabilization.

#### 3.3.2. Amplitude Sweep

Both oleogels and the NEs showed LVR at lower shear rates, followed by an exponential decrease and then the cross-over of G′ and G″ (Figure 6c,d). G′_LVR_ (Figure 6g) and the distance between G′ and G″ curves (Figure 6c,d) were found to be decreasing with the water fraction in NEs. Except for 10LStW90, all the NEs followed the same pattern. The presence of a shorter LVR length and a smaller distance between the G′ and G″ curves indicate that LSa NEs possess more rigidity compared to LSt NEs. The prevalent gelator junction zones in LSa emulsions have broken quickly at higher shear rates and led to shorter LVR compared to LSt emulsions. The sparse gelator networks in LSt oleogels and the corresponding NEs facilitated the prolonged reduction in elasticity. Interestingly, critical stress (C_s_) and yield stress (Y_s_) decreased with the increased water fraction in LSa emulsions; on the contrary, C_s_ and Y_s_ increased with the water fraction in LSt emulsions (Figure 6h). With the increase in water fraction, 10LSa-based NEs showed a consistent decrease in the rheological parameters (G′_LVR_, C_s_ and Y_s_) that define the strength of the material. On the other hand, 10LSt-based NEs showed inconsistency, which made it difficult to understand the influence of the addition of water to oleogels. The non-homogeneous and sporadic gelator fibers seem to be responsible for the inconsistency in the rheological properties of LSt-based gelled-oil NEs.

#### 3.3.3. Frequency Sweep

Predominant elastic gel-like behavior (G′ > G″) with minimal dependency on the angular frequency (Figure 6e,f) was found in all the oleogels and NEs. The G′ and G″ curves are almost parallel to each other and to the *X*-axis over the entire frequency range. The NEs with higher water volume showed a positive slope, especially for G″ curves in both kinds of NEs, indicating their increasing viscous nature. The nearly independent viscoelastic nature can be attributed to the high inter-particle interactions, which prevent the coalescence of dispersed gelled-oil particles and impart stability to the emulsions [34]. Similar results were reported for oleofoams and emulsions for cosmetic and food applications [35].

#### 3.3.4. Flow Ramp

A linear decrease in viscosity with the increasing shear rate without reaching the plateau indicates the power law relaxation mechanism (Appendix A). The flow curves exhibited a typical shear thinning behavior, with near constant viscosity at lower shear rates. The flow behavior of the NEs was understood by fitting the data (*η_app_*: apparent viscosity, *γ*: shear rate) in the modified power law model, $\eta_{app}=K.\gamma^{n-1}$. This model confirms that NEs exhibited pseudoplastic shear-thinning behavior. The flow behavior index (*n*) was found to be less than 1, confirming the non-Newtonian, pseudoplastic nature of the NEs (Appendix A). The consistency index (*K*) values indicated that NEs’ stability decreased with the increase in water fraction.

### 3.4. Loading of Curcumin

The solubility of curcumin in pure sunflower oil was found to be 3.10 ± 0.1 mg/g, and this aligns with the literature [36,37]. The solubility increased progressively with the concentration of gelators, reaching ~125% when LSt was 20% *w*/*w* (Figure 7a). The increment was also seen with LSa (~35%), but it was not as high as that of LSt. When St was added, curcumin forms a complex through both polar hydrogen bonds and non-polar interactions [27]. However, only hydrophobic interactions occur between Sa and curcumin. Because of the differences associated with these mechanisms, the maximum solubilized curcumin was 5.5 ± 0.3 and 4.0 ± 0.2 mg/g in 10LSt and 10LSa oleogels, respectively. Based on this, a uniform curcumin concentration (4 mg/g) was chosen for the encapsulation and in vitro release studies. The encapsulation efficiency (EE) of curcumin was in the order of 10LStW70 > 10LSaW70 > control (Table 2). The highest EE of LSt NEs is due to hydrogen bonds between the phenolic groups of curcumin and the hydroxyl groups of lecithin and St [38].

### 3.5. In Vitro Release of Curcumin

Curcumin release from the stable NEs (10LSaW70 and 10LStW70) and control (lecithin) was studied under in vitro gastric and small intestinal conditions. The control, a simple emulsion that does not have a consistent gelator network, showed the highest CPCR (cumulative percentage of curcumin release). After 3 h of gastric digestion, the observed CPCR from LSa, LSt and control emulsions was 7.72%, 9.86% and 85.4%, respectively (Figure 7b). Higher retention can be corroborated with the gelator network within the oil droplets and the interfacial gelator fibers. The steric hindrances and lipophilic activity of the gelator self-assemblies might have created a competitive environment for pepsin and prevented its adhesion on the oil droplets’ surface.

When the medium was changed to simulated intestinal fluid, more curcumin was released in the same fashion (control > LSt > LSa). Since ~95% of the curcumin was released within 4 h of digestion, further sampling was stopped with the control. After 6 h, the CPCR was 68.3%, and it was 75.4% from the 10LSaW70 and 10LStW70 emulsions, respectively (Figure 7b). Though Sa and St are stable at an intestinal pH, they are susceptible to pancreatin, as the enzyme can digest fatty acids [39]. However, after 9 h of digestion, ~32% and ~25% of curcumin still remained in the LSa and LSt emulsions, respectively. Pancreatin breaks lipids into small fat coacervates in the bolus [40,41]. In the present study, oleogels were also digested in a similar fashion, and curcumin was trapped within the undigested gelator networks of oil coacervates. The formation of such particles was confirmed by DLS. This was done after passing the in vitro digested samples through 0.45 µm PTFE (polytetrafluoroethylene) syringe filters. The size of the oleogel particles was significantly lower compared to the undigested samples (insert in Figure 7b). The release profiles and curcumin retention after digestion indicate that better entrapment of curcumin was achieved within the LSa and LSt networks of the NEs. The gelators’ networks acted as a physical barrier during curcumin release [12].

The effect of temperature on the gelator networks was studied by releasing curcumin at 25 and 50 °C in neutral PBS (pH 7). Under both temperatures, a higher rate of release was observed initially, and then a plateau was reached (Figure 7c,d). The initial high release can be attributed to the release of unentrapped curcumin or release from the phase-partitioned gelator fibers. The slow release of curcumin from the gelled-oil dispersion resulted in a plateau phase. A higher CPCR at 50 °C indicates that gelator networks are susceptible to high temperatures. The probable reasons for the elevated CPCR include destabilization of the gelator network and increased solubility of the curcumin at higher temperatures. The higher CPCR from 10LStW70 suggest its weaker thermal strength against 10LSaW70 (evident from temperature ramp studies). It was shown that both the emulsions started losing linear viscoelasticity at ~45 °C. This suggests that the emulsions might have destabilized at the operational temperature (50 °C). Though the NEs are thermosensitive, the gelator networks have not completely lost their structural integrity, evident from the sustained curcumin release at 50 °C. This is because of the higher melting points of Sa (69.3 °C) and St (53 °C) with respect to the operational temperature.

## 4. Conclusions

LSa and LSt oleogels were used as the precursors for preparing stable nanoemulsions (NEs) without additional surfactants. It was found that the self-assembled fibrillar networks (SAFiNs) have a dual function by structuring the dispersed oil and stabilizing the dispersion by stearic hindrance, performing a role similar to conventional emulsifiers and stabilizers. The presence of SAFiNs within the gelled-oil dispersions was confirmed by CLSM and rheological temperature ramp studies. The better performance of LSt was attributed to its sparsely distributed small fibers that result in more efficient self-assembled fibrillar networks. On the other hand, LSa formed large needle-like fibers.

The developed NEs demonstrated remarkable physical and thermal stability and were able to perform a controlled release of curcumin at three temperatures (25, 37 and 50 °C) relevant to beverage use. Since reducing the use of surfactants and stabilizers is one of the main challenges in food emulsions, the developed oleogel-based NEs can become a promising platform for the delivery of bioactive agents in beverages and other food applications.

## Figures and Tables

**Figure 1 foods-13-00680-f001:**
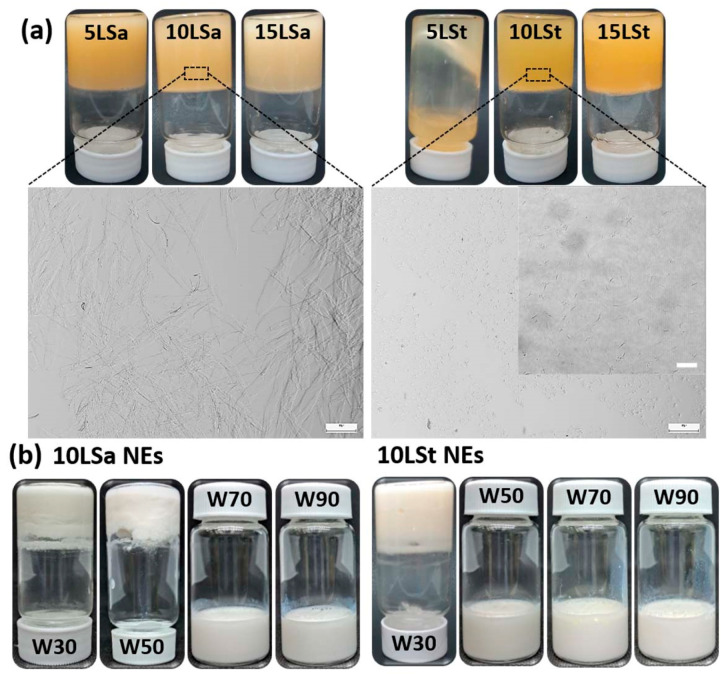
(**a**) Inverted vials and the microstructures of LSa and LSt oleogels. The scale bar size is 100 µm and 20 µm in the insert. (**b**) 10LSa- and 10LSt-based NEs with different water content.

**Figure 2 foods-13-00680-f002:**
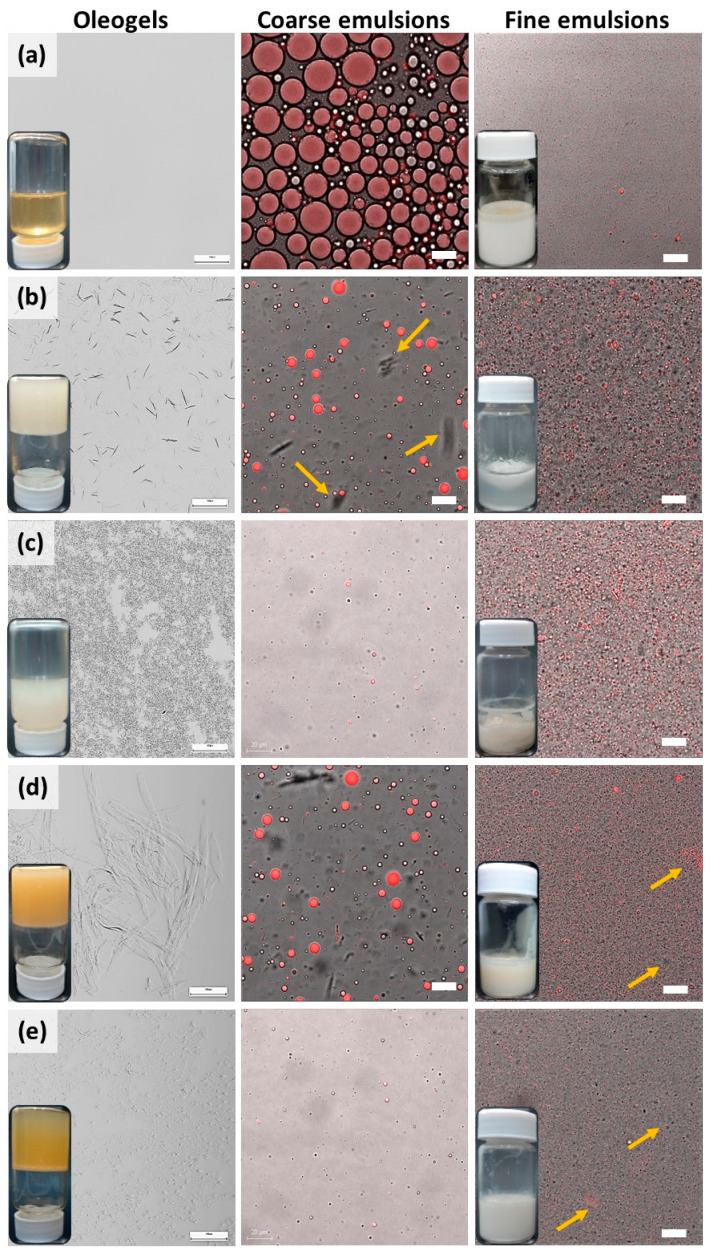
Micrographs and vials of oleogels and emulsions of (**a**) lecithin, (**b**) Sa, (**c**) St, (**d**) 10LSaW70 and (**e**) 10LStW70. Left to right: Light micrograph of each gelator is followed by confocal micrographs of coarse and fine emulsions. The scale bars in oleogels (left), coarse (middle) and fine (right) emulsions denote 100 µm, 20 µm and 10 µm, respectively. Microscopy revealed that 10LStW70 had better droplet dispersion compared to 10LSaW70.

**Figure 3 foods-13-00680-f003:**
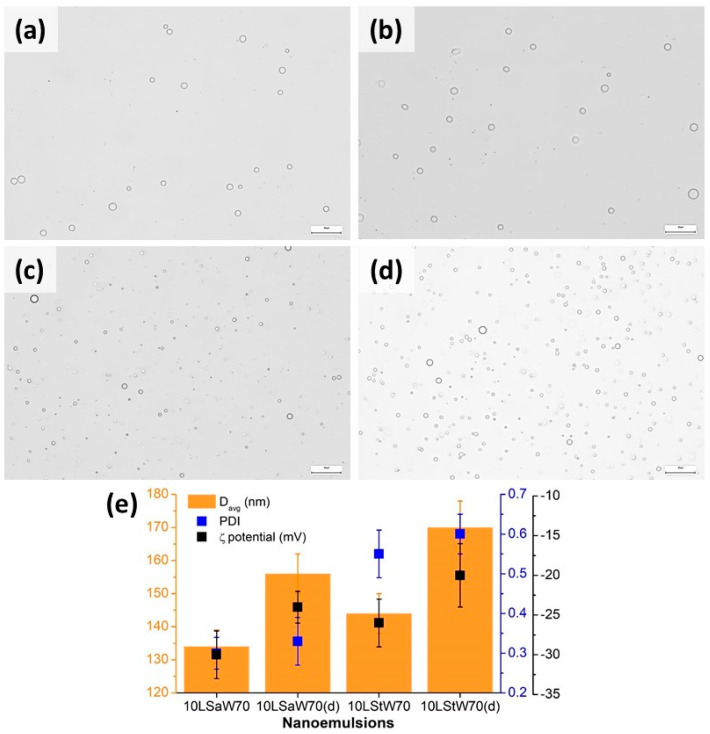
Micrographs of coarse emulsions of (**a**) 10LSaW70, (**b**) 10LSaW70 (diluted), (**c**) 10LStW70, (**d**) 10LStW70 (diluted) and (**e**) size, PDI and ζ-potential of NEs. The scale bars in the micrographs denote 20 µm.

**Figure 4 foods-13-00680-f004:**
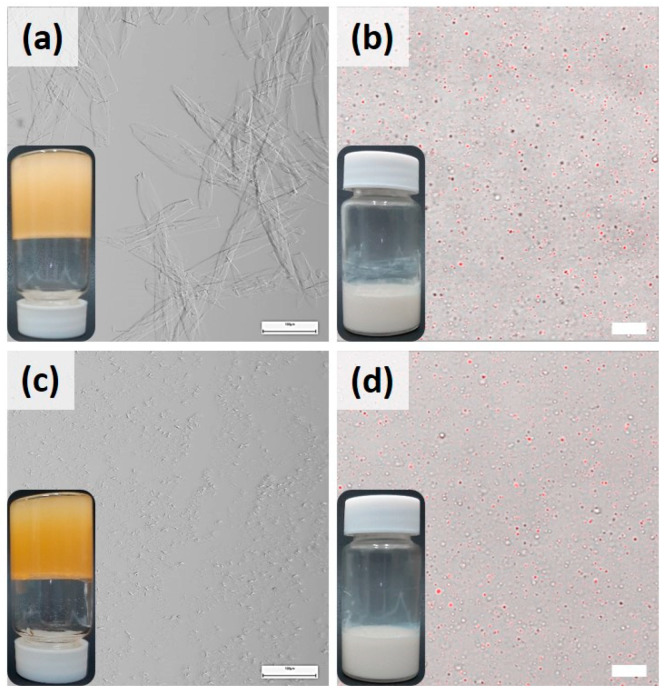
Light micrographs of (**a**) 10LSa and (**c**) 10LSt oleogels and confocal micrographs of (**b**) 10LSaW70 and (**d**) 10LStW70 NEs after 6 months of storage at 4 °C. The scale bars in oleogels (left), NEs (right) denote 100 µm and 10 µm, respectively.

**Figure 5 foods-13-00680-f005:**
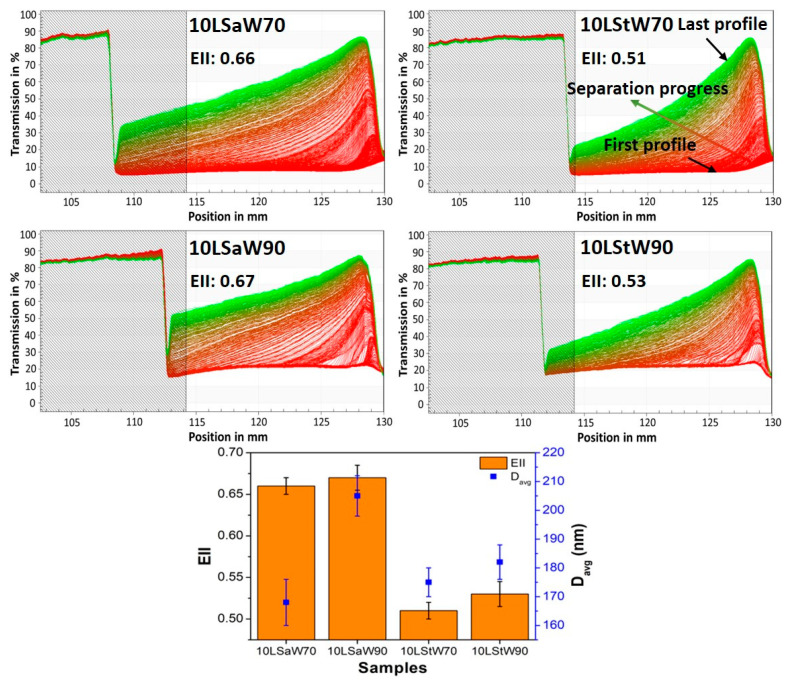
Lumisizer profiles of NEs along with their EII and D_avg_ of the dispersed phase.

**Figure 6 foods-13-00680-f006:**
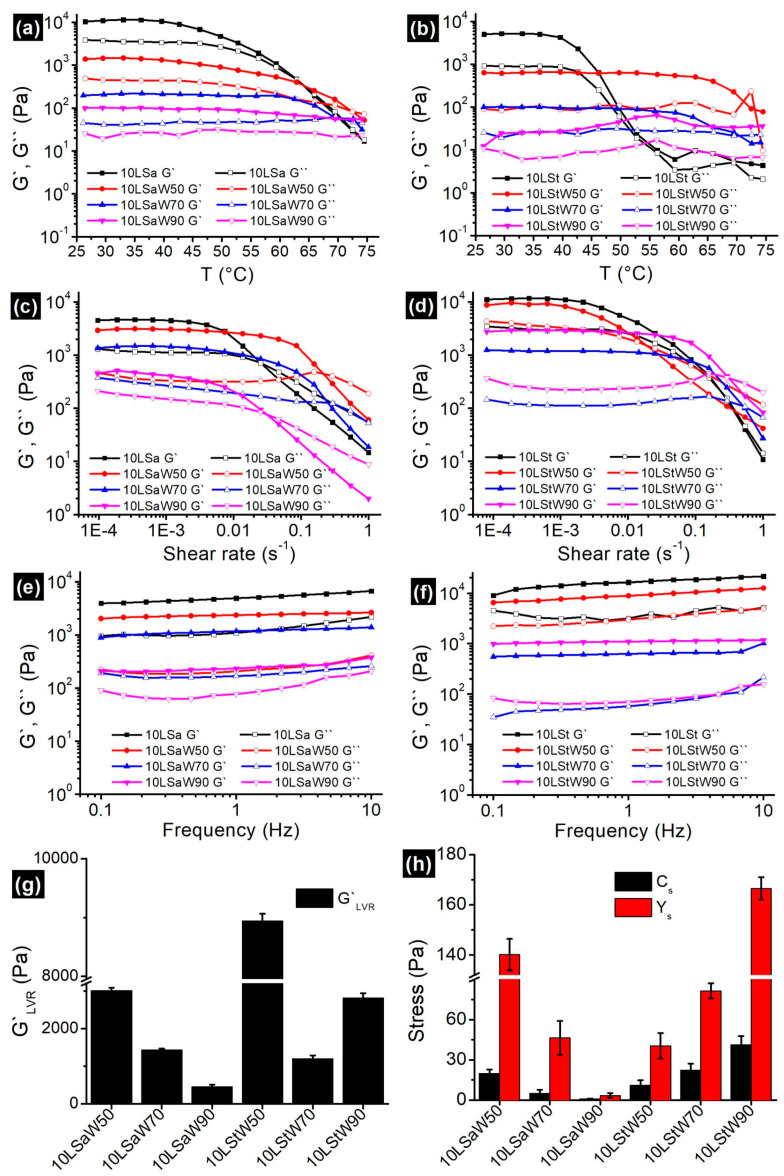
Change in G` and G`` of oleogels and NEs (**a**,**b**) as a function of temperature (**c**,**d**); amplitude sweep; (**e**,**f**) frequency sweep. The obtained (**g**) G`LVR and (**h**) Cs, Ys from the amplitude sweep studies are shown as bar graphs.

**Figure 7 foods-13-00680-f007:**
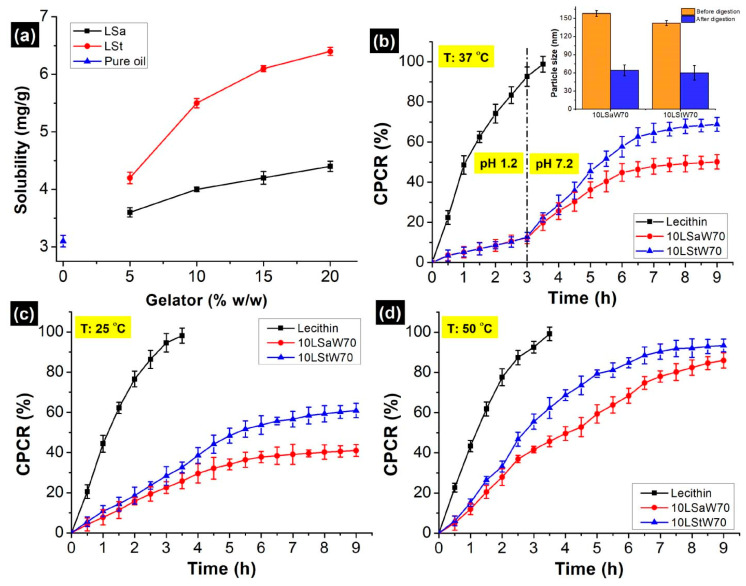
(**a**) Solubility of curcumin in sunflower oil and in molten oleogels containing variable amounts of gelators. In vitro release profiles of curcumin, cumulative percentage of curcumin release (CPCR) from stable NEs at (**b**) 37 °C, (**c**) 25 °C and (**d**) 50 °C.

**Table 1 foods-13-00680-t001:** Variables and their levels evaluated to obtain stable gelled-oil NEs.

Process and Formulation Variables	Levels
Gelator type and concentration	LSa: 5, 10, 15% (*w*/*w*) LSt: 5, 10, 15% (*w*/*w*)
Sonication amplitude	30, 50, 70%
Sonication time (min)	5, 10, 15
Oleogel/aqueous phase ratio % (*w*/*w*)	70/30, 50/50, 30/70 and 10/90

**Table 2 foods-13-00680-t002:** Physical characteristics of oleogel-based NEs: curcumin entrapment efficiency (EE), colloidal stability (CS) and physical state. 10LStW70 showed better performance than other stable emulsions.

Oleogel (Og)	Og/W Ratio	Emulsion Acronym	EE of Curcumin (%)	CS ^†^	Physical State
0% *	30/70		54.88 ± 2.66 ^a^	Not stable	Liquid
10LSa	10/90	10LSaW90	n.d	Stable	Liquid
	30/70	10LSaW70	76.62 ± 3.42 ^b^	Stable ^†^	Liquid
	50/50	10LSaW50	n.d	Creaming	Semi-solid
	70/30	10LSaW30	n.d	Oil syneresis	Semi-solid
10LSt	10/90	10LStW90	n.d	Stable	Liquid
	30/70	10LStW70	90.44 ± 2.76 ^c^	Stable	Liquid
	50/50	10LStW50	n.d	Stable	Liquid
	70/30	10LStW30	n.d	Oil syneresis	Semi-solid

^†^ The dispersion is considered colloidal stable when particles do not show a statistically significant size variation (*p* > 0.05) compared with the initial size. * Sunflower oil with lecithin. n.d = not determined. Mean ± standard deviation, means with different letters within the column are significantly different (*p* < 0.05).

## Data Availability

The original contributions presented in the study are included in the article/Appendix A, further inquiries can be directed to the corresponding author.

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
