# Peer review of "Oleogel-Based Nanoemulsions for Beverages: Effect of Self-Assembled Fibrillar Networks on Stability and Release Properties of Emulsions"

_foods, 2024, doi:10.3390/foods13050680_

Round 1

Reviewer 1 Report

Comments and Suggestions for Authors

In this manuscript, microscopy and rheometry revealed that the presence of self-assembled fibrous networks (SAFiNs) in both dispersed and continuous phases provided steric stabilization to NEs. Lecithin acted as crystal habit modifier of SAFiNs and facilitated their phase partitioning. Notably, short fibers of LSt showed better emulsifying efficiency than the long fibers of LSa. Curcumin release studies under simulated gastrointestinal conditions demonstrated that SAFiNs affect the release capabilities of NEs. Polydispersity index, zeta potential and oil syneresis data showed that emulsions are stable for six months. Moreover, NEs showed thermal stability upon curcumin release at 25 and 50 °C. These results suggested that the developed oleogel-based NEs are suitable for delivery of bioactive agents for beverages and other food applications. However, there are still some issues that need to be revised in this article:

1.       Lien number is necessary.

2.       In introduction part. The delivery system should be introduced, please refer this reference(Journal of Agricultural and Food Chemistry, 70(21):6300-6316).

3.       Oleogel-based nanoemulsions should be introduced.

4.        “2.2. Characterization of oleogel-based nanoemulsions”. Please refer this reference (Food Research International, 175(2024), 113726.).

5.       The significant difference should be analyzed in the Table and Figure.

6.       The reference should be updated in recent years.

Comments on the Quality of English Language

In this manuscript, microscopy and rheometry revealed that the presence of self-assembled fibrous networks (SAFiNs) in both dispersed and continuous phases provided steric stabilization to NEs. Lecithin acted as crystal habit modifier of SAFiNs and facilitated their phase partitioning. Notably, short fibers of LSt showed better emulsifying efficiency than the long fibers of LSa. Curcumin release studies under simulated gastrointestinal conditions demonstrated that SAFiNs affect the release capabilities of NEs. Polydispersity index, zeta potential and oil syneresis data showed that emulsions are stable for six months. Moreover, NEs showed thermal stability upon curcumin release at 25 and 50 °C. These results suggested that the developed oleogel-based NEs are suitable for delivery of bioactive agents for beverages and other food applications. However, there are still some issues that need to be revised in this article:

1.       Lien number is necessary.

2.       In introduction part. The delivery system should be introduced, please refer this reference(Journal of Agricultural and Food Chemistry, 70(21):6300-6316).

3.       Oleogel-based nanoemulsions should be introduced.

4.        “2.2. Characterization of oleogel-based nanoemulsions”. Please refer this reference (Food Research International, 175(2024), 113726.).

5.       The significant difference should be analyzed in the Table and Figure.

6.       The reference should be updated in recent years.

Author Response

Dear Reviewer,

We would like to thank you for reviewing our manuscript foods-2864129 entitled “Oleogel-based nanoemulsions for beverages: Effect of self-assembled fibrillar networks on stability and release properties of emulsions” by Sai Sateesh Sagiri and Elena Poverenov. We have carefully considered all the reviewers' comments and made necessary corrections and changes. All the changes in the revised manuscript are highlighted in yellow. The list of our responses to the reviewer's comments follows this cover letter. To provide point-to-point responses, we have embedded our responses (blue) in the original text of the reviewer's comments.

Reviewer #1:

In this manuscript, microscopy and rheometry revealed that the presence of self-assembled fibrous networks (SAFiNs) in both dispersed and continuous phases provided steric stabilization to NEs. Lecithin acted as crystal habit modifier of SAFiNs and facilitated their phase partitioning. Notably, short fibers of LSt showed better emulsifying efficiency than the long fibers of LSa. Curcumin release studies under simulated gastrointestinal conditions demonstrated that SAFiNs affect the release capabilities of NEs. Polydispersity index, zeta potential and oil syneresis data showed that emulsions are stable for six months. Moreover, NEs showed thermal stability upon curcumin release at 25 and 50 °C. These results suggested that the developed oleogel-based NEs are suitable for delivery of bioactive agents for beverages and other food applications. However, there are still some issues that need to be revised in this article:

  1. Line number is necessary.

            As per the reviewer's suggestion, line numbers are provided.

  1. In introduction part. The delivery system should be introduced, please refer this reference (Journal of Agricultural and Food Chemistry, 70(21):6300-6316).

            As per the suggestion, the ability of Oleogels and oleogel-based emulsions as delivery system was introduced with appropriate references in page 2, line 18.

“The wide range of oleogels and oleogel-based emulsions applications can also be attributed to their ability to release the active ingredients (e.g. drugs, antioxidants, nutraceuticals) in controlled manner (18, 19).”

  1. Pinto TC, Martins AJ, Pastrana L, Pereira MC, Cerqueira MA. Oleogel-Based Systems for the Delivery of Bioactive Compounds in Foods. Gels. 2021 Sep;7(3):86.
  2. Zhang Y, Dong L, Liu L, Wu Z, Pan D, Liu L. Recent Advances of Stimuli-Responsive Polysaccharide Hydrogels in Delivery Systems: A Review. J Agric Food Chem. 2022 Jun 1;70(21):6300–16.

  1. Oleogel-based nanoemulsions should be introduced.

            As per the suggestion, oleogel-based nanoemulsions was introduced at page 2, line 25.

“Previously, oleogel-based NEs were designed as gelled oil particles using fats (monoglycerides) and composite gelators (carnauba wax/1-docosonol, ovalbumin/gum arabic) (21-23). In this study, sunflower oil was structured using biocompatible gelators, lecithin (L) in combination with simple fatty acids such as stearic acid (Sa) or sorbitan tristearate (St).”

  1. Palla CA, Aguilera-Garrido A, Carrín ME, Galisteo-González F, Gálvez-Ruiz MJ. Preparation of highly stable oleogel-based nanoemulsions for encapsulation and controlled release of curcumin. Food Chemistry. 2022 Jun 1;378:132132.
  2. Xia T, Wei Z, Xue C. Impact of composite gelators on physicochemical properties of oleogels and astaxanthin delivery of oleogel-based nanoemulsions. LWT. 2022 Jan 1;153:112454.
  3. Gao Y, Wang Z, Xue C, Wei Z. Modulation of Fabrication and Nutraceutical Delivery Performance of Ovalbumin-Stabilized Oleogel-Based Nanoemulsions via Complexation with Gum Arabic. Foods. 2022 Jan;11(13):1859.

  1. “2.2. Characterization of oleogel-based nanoemulsions”. Please refer this reference (Food Research International, 175(2024), 113726.).

            We thank the reviewer for improving the quality of the manuscript. The suggested reference was used at Page 4.

  1. The significant difference should be analyzed in the Table and Figure.

            In Table 2 and Figure 2 legends, the observed significant difference between the stable emulsions was written.

“Table 2: 10LStW70 showed better performance than other stable emulsions.

Figure 2: Microscopy revealed that 10LStW70 has better droplets dispersion compared to 10LSaW70.”

  1. The reference should be updated in recent years.

            As the reviewer suggested, the references were updated. The references earlier to 2014 were replaced with recent references.

  1. Nasrolahi S, Sadeghizadeh-Yazdi J, Ehrampoush MH, Madadizadeh F, Khalili E. Evaluation of rheological and optical properties plus stability of beverage cloud emulsions prepared with corn oil, gum rosin, and modified starch. Food Science & Nutrition. 2023;11(2):806–16.
  2. Pinto TC, Martins AJ, Pastrana L, Pereira MC, Cerqueira MA. Oleogel-Based Systems for the Delivery of Bioactive Compounds in Foods. Gels. 2021 Sep;7(3):86.
  3. Palla CA, Aguilera-Garrido A, Carrín ME, Galisteo-González F, Gálvez-Ruiz MJ. Preparation of highly stable oleogel-based nanoemulsions for encapsulation and controlled release of curcumin. Food Chemistry. 2022 Jun 1;378:132132.
  4. Xia T, Wei Z, Xue C. Impact of composite gelators on physicochemical properties of oleogels and astaxanthin delivery of oleogel-based nanoemulsions. LWT. 2022 Jan 1;153:112454.
  5. Gao Y, Wang Z, Xue C, Wei Z. Modulation of Fabrication and Nutraceutical Delivery Performance of Ovalbumin-Stabilized Oleogel-Based Nanoemulsions via Complexation with Gum Arabic. Foods. 2022 Jan;11(13):1859.
  6. Chen Q, Liu Y, Li Y, Dong L, Liu Y, Liu L, et al. Interaction and binding mechanism of ovalbumin with cereal phenolic acids: improved structure, antioxidant activity, emulsifying and digestion properties for potential targeted delivery systems. Food Research International. 2024 Jan 1;175:113726.

            In addition, the manuscript was checked carefully, and the mistakes were corrected if found any. We thank the reviewer for providing the comments and improving the quality of the manuscript for possible publication in the journal.

Reviewer 2 Report

Comments and Suggestions for Authors

 Reducing the use of stabilizers in beverages is currently a challenge.

The aim presented by the authors of the study was to produce a nanoemulsion with an oleogel structure (NE) without the use of surfactants. Lecithin-stearic acid and lecithin-sorbitan tristearate oleogels were used as precursors to prepare stable nanoemulsions (NE) without additional surfactants. Lecithin-stearic acid and lecithin-sorbitan tristearate oleogels formed stable NE under optimized sonication conditions. It was found that self-assembling fibrillar networks (SAFiN) provide structure to the dispersed oil and stabilize it, playing a similar role to the emulsifiers and stabilizers used. Among the many desirable properties of the obtained nanoemulsion, the important ones include the stability of the emulsion for six months and thermal stability after the release of curcumin at temperatures of 25 and 50 ° C. The comprehensive results of testing the obtained emulsion presented by the authors suggest that the developed oleogel-based NEs are suitable for delivering bioactive agents to beverages. They may have wider applications in the food industry.   The work was edited correctly, the introduction to the topic was sufficient. Materials and methods described correctly. Please correct "thermos scientific" and "Agene international" to the correct company names. Please write Na2HPO4 and KH2PO4 correctly with subscripts respectively. Please separate the degrees C in the superscript, it appears several times in the text. You should put clear scales on FIG. 1, 2, 3. The font of Fig. 2 is different from Fig. 1 and Fig. 3 On page 11, last line, please correct the "dashes" before mV. Statistically significant differences are included in table 1, there are no graphs, was it intentional? If not, please plot the statistical differences on the graphs.  

Interesting article, appropriately selected methods, reliable characterization of the obtained emulsion.

Author Response

Dear Reviewer,

We would like to thank you for reviewing our manuscript foods-2864129 entitled “Oleogel-based nanoemulsions for beverages: Effect of self-assembled fibrillar networks on stability and release properties of emulsions” by Sai Sateesh Sagiri and Elena Poverenov. We have carefully considered all the reviewers' comments and made necessary corrections and changes. All the changes in the revised manuscript are highlighted in yellow. The list of our responses to the reviewer's comments follows this cover letter. To provide point-to-point responses, we have embedded our responses (blue) in the original text of the reviewer's comments.

Reviewer #2:

Reducing the use of stabilizers in beverages is currently a challenge.

The aim presented by the authors of the study was to produce a nanoemulsion with an oleogel structure (NE) without the use of surfactants. Lecithin-stearic acid and lecithin-sorbitan tristearate oleogels were used as precursors to prepare stable nanoemulsions (NE) without additional surfactants. Lecithin-stearic acid and lecithin-sorbitan tristearate oleogels formed stable NE under optimized sonication conditions. It was found that self-assembling fibrillar networks (SAFiN) provide structure to the dispersed oil and stabilize it, playing a similar role to the emulsifiers and stabilizers used. Among the many desirable properties of the obtained nanoemulsion, the important ones include the stability of the emulsion for six months and thermal stability after the release of curcumin at temperatures of 25 and 50 °C. The comprehensive results of testing the obtained emulsion presented by the authors suggest that the developed oleogel-based NEs are suitable for delivering bioactive agents to beverages. They may have wider applications in the food industry. The work was edited correctly, the introduction to the topic was sufficient. Materials and methods described correctly.

  1. Please correct "thermos scientific" and "Agene international" to the correct company names.

We thank the reviewer for correcting the company names. The company names were corrected to “thermo scientific, Angene international Ltd” in Page 3, line 16.

  1. Please write Na2HPO4 and KH2PO4 correctly with subscripts respectively.

We thank the reviewer for improving the quality of the manuscript. Na2HPO4 and KH2PO4 were rewritten using subscripts in Page 3, line 18 as Na2HPO4, KH2PO4.

  1. Please separate the degrees C in the superscript, it appears several times in the text.

We beg to differ from the opinion of the reviewer about describing the degrees C. For e.g. The scientific representation of five degrees C is 5 °C not 5° C. Degrees C should not be separated with space.

  1. You should put clear scales on FIG. 1, 2, 3. The font of Fig. 2 is different from Fig. 1and Fig.3

We agree with the reviewer concern upon the clarity and font of scales in the figures. Since the scales were embedded in the images while photographing, we could not edit them appropriately. Henceforth, for better understanding, the scale of the size bars was clearly mentioned in the legends of Fig. 1 and 2. We made a mistake of not mentioning the size of the scale bar in Fig. 3. The mistake was rectified by writing “The scale bars in the micrographs denote 20 µm” in the legend of Fig. 3 (Page 12, line 9).

  1. On page 11, last line, please correct the "dashes" before mV.

We thank the reviewer for correcting the mistake. The ζ-potential values were represented as –18 to –32 mV for all the concentrated and diluted emulsions and –29 to –16 mV for stable emulsions after six months of storage (Page 13, line 1).

  1. Statistically significant differences are included in table 2, there are no graphs, was it intentional? If not, please plot the statistical differences on the graphs.

We thank the reviewer for the suggestion. As the statistical difference between the encapsulation efficiency of curcumin in emulsions was clearly seen with 54%, 76% and 90% for lecithin emulsions, 10LSaW70 and 10LStW70, respectively, separate statistics graph was not drawn.

Interesting article, appropriately selected methods, reliable characterization of the obtained emulsion.

We thank the reviewer for the valuable suggestions and appreciating the significance of the article for possible publication.
